# Using orthogonal vectors to improve the ensemble space of the EnKF and its effect on data assimilation and forecasting

Yung-Yun Cheng[1], Shu-Chih Yang[2,3], Zhe-Hui Lin[2], and Yung-An Lee[2]

[1]Center for Weather Climate and Disaster Research, National Taiwan University, Taipei, Taiwan
[2]Department of Atmospheric Sciences, National Central University, Taoyuan, Taiwan
[3]GPS Science and Application Research Center, National Central University, Taoyuan, Taiwan

*Correspondence to*: Dr. Shu-Chih Yang (shuchih.yang@atm.ncu.edu.tw)

**Abstract.** The space spanned by the background ensemble provides a basis for correcting forecast errors in the ensemble Kalman filter. However, the ensemble space may not fully capture the forecast errors due to the limited ensemble size and
systematic model errors, which affect the assimilation performance. This study proposes a new algorithm to generate pseudo members to properly expand the ensemble space during the analysis step. The pseudomembers adopt vectors orthogonal to the original ensemble and are included in the ensemble using the centered spherical simplex ensemble method. The new algorithm is investigated with a six-member ensemble Kalman filter implemented in the Lorenz 40-variable model. Our results suggest that the ensemble singular vector, ensemble mean vector and their orthogonal components can serve as effective pseudo
members for improving the analysis accuracy, especially when the background has large errors.

## 1 Introduction

The ensemble Kalman filter (EnKF) has the great advantage of using flow-dependent background error covariance (BEC) and has been widely applied to state estimation in geophysics. The BEC is estimated by the background ensemble, and its characteristic is crucial since it determines how the observations are spread out to correct the model state. The space spanned
by the background ensemble members (the ensemble space) is expected to capture the dynamically growing errors, and the ensemble space provides a basis for corrections.

However, the use of a finite ensemble size can cause the underestimation of background error variance, and the background error correlation is less optimally represented due to the sampling error. Therefore, the growing errors may not be well captured, and corrections from the EnKF are less optimal. Bocquet and Carrassi (2017) indicated that the stability of the EnKF relies on
the subspace spanned by the ensemble members that represent the unstable-neutral subspace. In other words, maintaining the ensemble space is important for the EnKF performance. Strategies such as additive covariance inflation (Whitaker et al., 2008) or hybrid methods (Hamill and Snyder, 2000) are commonly used to increase the dimensionality of the ensemble. These methods expand the overall ensemble space but are operated empirically without a particular direction.

Previous studies have suggested that vectors stimulate the growing modes can improve the dimensionality of the ensemble
space and the performance of the EnKF. For example, Carrassi et al. (2008) used bred vectors as the direction of the analysis increment to update the analysis states in an unstable area. Yang et al. (2015) used a two-sided method to apply the initial

ensemble singular vectors (IESV) as additive inflation to correct the fastest-growing errors in a quasigeostrophic model. Chang et al. (2020), under the framework of a hybrid-gain data assimilation framework (Penny, 2014), used part of the variational information orthogonal to the EnKF analysis perturbation to correct the EnKF means. These studies emphasize the importance

of generating additional effective correction. Inspired by these works, this study proposes generating pseudoensemble members to increase the ensemble space to better capture forecast errors without increasing the computational cost. We investigate whether the use of pseudomembers could improve the analysis and forecast and which type of pseudomember is most effective in this regard.

This paper is organized as follows. Section 2 introduces the generation of pseudomembers. Section 3 presents the impact of

40 using pseudomembers for EnKF analysis. Section 4 provides the summary and discussion of this work.

## 2 Methodology and experimental design

### 2.1 General setup

This study conducts a series of observation simulation system experiments (OSSEs) with the local ensemble transform Kalman filter (LETKF; Hunt et al., 2007) implemented in the 40-variable Lorenz-96 model (Lorenz 1996, Lorenz and Emanuel 1998)

(See Appendix A for the detailed procedure). The standard set of experiments performs 6-member LETKF every 30 steps with observations available every two grid points. Table 1 lists the details of the experiments and assimilation configurations.

Before performing data assimilation, a new vector is included as the extra ensemble member. The traditional double-sized method (Toth and Kalnay 1993) requires an even number of included members and can lead to ill-conditioned problems during EnKF computation. Therefore, we adopt the centered spherical simplex ensemble (CSSE; Wang et al., 2004) method, which

can add any number of members without modifying the ensemble mean and spread. More importantly, the CSSE method avoids ill-conditioned problems.

### 2.2 Deriving the vectors for pseudomembers

An added member is referred to as a pseudomember given that it is generated at the analysis time and is not used during the forecast stage. Two types of vectors are used for generating pseudomembers, including the initial ensemble singular vector

(IESV) and ensemble mean vector (EMV). Given a set of ensemble forecasts, IESV finds fast-growing perturbations within a period by linearly combining the ensemble perturbations (Enomoto et al., 2015, Yang et al., 2015). In ensemble data assimilation, EMV is used to define the ensemble perturbations (as the deviation) and is not accounted for in the degrees of freedom, although the perturbations evolve upon the mean state. However, it is likely that the forecast errors carry a component with the structure of the mean, such as the large-scale pattern, and will not be represented in the ensemble perturbations.

With the generated vector, we further use its component orthogonal to the ensemble space added as the pseudomember for EnKF computation. We first apply SVD to find the orthogonal vectors to represent the space spanned by the ensemble

members. Second, we use Equations (1) and (2) to obtain the orthogonal component of the generated vector (orthogonal IESV1 or orthogonal EMV).

$$\mathbf{v}_{\text{final proj}} = \sum_{i=1}^{K-1} \frac{(\tilde{\mathbf{v}} \cdot \mathbf{v}_i)}{|\mathbf{v}_i|^2} \mathbf{v}_i \tag{1}$$

$$\mathbf{v}_{\text{orth}} = \text{normalize}(\tilde{\mathbf{v}} - \mathbf{v}_{\text{final proj}}) \tag{2}$$

where $\tilde{\mathbf{v}}$ is the normalized generated vector, $\mathbf{v}_i$ is the $i^{\text{th}}$ orthogonal vector of the ensemble space, $\mathbf{v}_{\text{final proj}}$ is the total projection of the generated vector at all the orthogonal vectors of the ensemble space, and $\mathbf{v}_{\text{orth}}$ is the orthogonal component of the generated vector. Finally, the orthogonal component of the generated vector is taken as the new ensemble perturbation for expanding the ensemble space.

The orthogonal vector is rescaled to have the amplitude of the background ensemble spread. Different experiments are designed. The control experiment (CNTL) conducts standard LETKF assimilation with six members and is taken as the baseline. We conduct four experiments with the added pseudomembers. The first two experiments use the global IESV1 and EMV, and the last one adds two orthogonal vectors from IESV1 and EMV.

## 2. 3 Setup of the increased-size EnKF system

Figure 1 shows the flow chart of our experiments. With the $K$-member background ensemble, $M$-member pseudovectors are generated. With CSSE, the ensemble size becomes $(K+M)$, and the LETKF analysis is performed with the new ensemble. We conducted our experiments with offline and online frameworks. The offline framework, in which the LEKTF analysis is not cycled, is used to investigate how the ensemble space varies after adding the pseudovector and to understand the benefits of the increased-size EnKF system by clean comparisons. The background ensemble is provided by the background ensemble of the CNTL at each analysis step. In contrast, the analysis is cycled in the online experimental framework to evaluate the accumulated feedback from the increased-size EnKF system. However, $K$ members need to be selected from the new $(K+M)$ members so that the following ensemble forecast is done without the need for extra computational costs. To do so, we remove the last $M$ members with Equation 3 to keep the ensemble mean and the ensemble spread the same as when using the $(K+M)$ members.

$$\mathbf{v}_i^{new} = \bar{\mathbf{v}}_i|_{(K+M)} + \left( \frac{1}{K} \sum_{j=K+1}^{K+M} \mathbf{v}'_j + \mathbf{v}'_i \right) \left( \frac{\sigma_{K+M}}{\sigma_K} \right) \quad i = 1,2 \dots K \tag{3}$$

where $\mathbf{v}_i^{new}$ is the new $i^{\text{th}}$ member, $\bar{\mathbf{v}}_i|_{(K+M)}$ is the mean of the $(K+M)$ members, $\mathbf{v}'_i$ is the $i^{\text{th}}$ member perturbation, and $\sigma_{K+M}$ and $\sigma_K$ represent the ensemble spread of the $(K+M)$ and $K$ members, respectively. In Equation (3), $\frac{1}{K} \sum_{j=K+1}^{K+M} \mathbf{v}'_j$ is used to ensure that the sum of the new perturbations of the first $K$ members is equal to zero (Appendix B).

 **3 Results**

This subsection illustrates how including the pseudomember modifies the space spanned by ensemble members. The results are obtained from the offline setting, in which the orthogonal vector is directly included in the background ensemble of the standard LETKF experiment (CNTL) without cycling the impact. In this study, the orthogonal vector is computed and added globally and can modify the BEC whose structure determines the analysis correction. Here, we focus on the local maximum

forecast error (LME) area, which is defined as a local area spanning seven grids and has the largest forecast error at its center. The characteristics of the ensemble space in the LME area are represented by the eigen spectrum of the local ensemble perturbation. Figure 2 shows the percentage of eigenvalues averaged for 550 DA cycles. The ensemble space of CNTL (black) has 5 nonzero eigenvalues with 6 members. The newly added orthogonal vector successfully provides an independent mode into the ensemble space. With two orthogonal vectors, the 8-member (Fig. 2, blue) experiment (Orth_EMV+Orth_IESV1) can

increase two independent modes, so the ensemble space is expanded into seven modes.

The expansion of the ensemble space can further modify the analysis correction. This can be illustrated by the projection of forecast errors onto the orthogonal vectors of the ensemble space and the reduction in the analysis errors with the additional orthogonal vector in the LME area. The calculation of the projection is similar to Eq. 2, except $\tilde{\mathbf{v}}$ is replaced with the error of the background ensemble mean and the amplitude of $\mathbf{v}_{\text{final proj}}$ is calculated at each analysis cycle in each experiment. We

compare how well the modification of the ensemble space can help to capture the error of the background ensemble mean. The projection of CNTL decreases when the LETKF performs poorly in reducing the background error (black line in Fig. 3b vs. Fig. 3a). This also confirms that LETKF assimilation is less successful when the ensemble cannot capture the forecast error well. All experiments with the additional orthogonal vector successfully increase the projection and provide a more effective correction to reduce the error (Fig. 3c), especially at the analysis times when the CNTL analysis errors are larger than the

background errors (highlighted by the red dashed boxes in Fig. 3). Furthermore, Orth_EMV+Orth_IESV1 has the best performance in capturing forecast errors and reducing analysis errors. Note that the projection of the forecast errors only illustrates how well the ensemble space encompasses the forecast errors. It should be noted that the forecast errors in the LME area, spanning 7 grids, can be fully represented by 7 orthogonal vectors (Orth_EMV+Orth_IESV1), but the background errors may not be completely removed due to issues such as the underrepresentation of background error variance. Nevertheless, the

offline experiments confirm the potential of adding the orthogonal vector to provide more effective corrections, and the improvement is highly flow dependent.

Figure 4 sorts the projection of the CNTL background ensemble on the background error according to the CNTL analysis RMSE and the projection of the added orthogonal vector on the forecast error residual. The forecast error residual is the unexplained forecast error after removing the projection of the original background ensemble from the original forecast

error. All calculations are performed on the whole domain. First, large CNTL analysis RMSE corresponds to the low projection of the original background ensemble on forecast errors (i.e., blue dots with large RMSEs). In general, the larger the CTRL RMSE is, the higher the orthogonal vector capturing the residual error. Compared to the orthogonal IESV1, the orthogonal

EMV projects more onto the forecast error residual at most analysis times. Therefore, the orthogonal EMV is more useful to increase the ensemble space to reduce the analysis errors in this example.

In this subsection, we compare the results of the online experiments, in which the impact of using the additional pseudovector is cycled during the analysis step and further feedbacks into the next background ensemble through analysis cycling. We evaluate the analysis error based on the RMSE of the analysis ensemble mean and the experiments with the standard configuration are repeated 10 times with different initial random seeds to show the statistical robustness. Based on Fig. 3, Table 2 focuses on the analysis cycles, whose analysis RMSE values in the CNTL are two standard deviations larger

than the mean CNTL RMSE. This highlights the effect of adding the new vectors in LETKF assimilation when the original background ensemble is less capable of reducing forecast errors. With 10 randomly initialized standard experiments, adding IESV1 or EMV as the pseudovector is always effective in improving the CNTL analysis. When the CTRL has large analysis errors, both IESV1 and EMV can have a mean improvement rate larger than 46%. This indicates that the additional correction is beneficial for correcting the growing error and thus results in positive feedback. On average, the Orth_EMV shows a larger

mean improvement than the EMV. It should be noted that adding the Orth_IESV1 always has a better performance than CNTL and sometimes than the IESV1. However, the Orth_IEVS1 may overly expand the subgrowing direction of the ensemble space, and the overcorrection results in a poorer performance than the IESV1 experiments. Therefore, the Orth_IESV1 has a smaller improvement rate than the IESV on average and the standard deviation of RMSE is much larger than that of the IESV1.

Table 3 summarizes the performance of the 6-member experiments with different assimilation configurations and

140 experiments are conducted using one of the initial conditions used in the CTRL experiments. Our results confirm that adding the orthogonal vector is always beneficial in improving the analysis accuracy under different assimilation configurations, including larger observation error, sparse observations, and short or long assimilation intervals. We further compare experiments using one or two orthogonal vectors with the standard LETKF with 7 members. While using more members introduces more computation during the ensemble forecast, experiments using the pseudovectors for assimilation do not have

145 extra computation for performing ensemble forecasting. Orth_EMV+Orth_IESV1 successfully combines the advantages of using the orthogonal IESV1 and the orthogonal EMV, and thus, it has better performance than both single-pseudomember experiments, especially for groups with large analysis errors (Fig. 5b,c). More importantly, Orth_EMV+Orth_IESV1 outperforms in the group with mildly large analysis errors (Fig. 5b). However, when the analysis error grows to a certain range, the 7-member standard LETKF (M7_LETKF) has the best performance. Such an improvement with pseudomembers is still

valid and even more evident when the model is imperfect (Fig. 5d-f), and Orth_EMV+Orth_IESV1 has a comparable performance with M7_LETKF in general (Fig. 5d).

We further justify this method with a large ensemble size (20 members) and investigate how many pseudomembers can be useful. As shown in Table 3, the benefit of adding pseudomembers based on IESVs increases as the number of pseudomembers increases, particularly when the standard LETKF (M20_CNTL) has a large analysis RMSE. Using more than

155 10 IESVs can reduce the analysis RMSE by 45%, and the IESV1 (M20_Orth_IESV1) provides the dominant effect. The improvement rate with 15 IESVs saturates for the following 30-step forecast. Such a performance (M20_Orth_IESV1-15) is

better than the 25-member standard LETKF in general and even better than the 30-member standard LETKF for the group of mildly large analysis errors (Figure 6). The forecast computation is only 66% of the 30-member LETKF. This also confirms that IESV-based pseudomembers can effectively expand the ensemble space to capture the growing forecast errors. We also note that M20_Orth_EMV is less effective than M20_Orth_IESV1 in the case of a large ensemble. The reason that the EMV as the pseudomember is very effective with the small ensemble is because the ensemble space spanned by the ensemble perturbations cannot capture the background error well, and the corrections for the background mean can be less optimal. This limitation is more evident for the large-scale error due to using a small localization. As a result, the structure of the ensemble mean largely projects on the background error (Fig. 4), and thus using the EMV as the pseudomemer leads to a good performance. With a large ensemble size and a large localization, the large-scale error in the background mean state is much reduced, and thus EMV is less effective in being used as the pseudomember. In comparison, the ensemble forecast can better capture the error evolution with more members, leading to a more robust IESV1. Nevertheless, including the orthogonal EMV is still more beneficial than using all pseudomembers with IESVs (M20_Orth_EMV+IESV1-4 vs. M20_Orth_IESV1-5).

## 4 Summary and conclusion

This study proposed a new idea of adding cost-free pseudomembers to expand the ensemble space and to improve the performance of EnKF assimilation and forecast. Based on the Lorenz-96 model, this idea is investigated with offline and online frameworks. Two types of pseudoensemble members are compared, including the global IESV1 and EMV. Both are very effective in expanding the ensemble space in sensitive areas while the orthogonal EMV is most effective in improving the analysis accuracy with a small ensemble size. With a large ensemble size, the structure of the mean state is less effective as the pseudomember, compared to IESV1. The effective improvement with IESV1 indicates the importance of maintaining the direction of growing error in the ensemble space. We also confirm that adding more pseudomembers with EMV and IESVs can further improve the analysis and forecast accuracy.

In the current operational ensemble DA system, additive covariance inflation is commonly adopted to improve the EnKF performance. However, how to choose additive inflation is ad hoc and may even break the balance of the structure of the original ensemble space and degrade the performance of the data assimilation system. The newly proposed simple idea could be a gentle alternative. In the future, we plan to explore the feasibility of this idea in a real model; such as using the orthogonal EMV as pseudomember for ocean ensemble data assimilation.

## Appendix A: Introduction of Lorenz-96 model

The Lorenz-96 model has been used to study the issue of error growth and the probability of atmosphere and weather forecasting (Kekem et al. 2018). The governing equation of the Lorenz-96 model is:

$$\frac{dx_j}{dt} = x_{j-1}\left(x_{j+1} - x_{j-2}\right) - x_j + F \qquad j = 1, \dots, n \qquad (A.1)$$

where $n$ is the dimension of the system, j is the index of the analysis grids, and $F$ is the external forcing term. $\frac{dx_j}{dt}$ can be interpreted as some atmospheric quantity measured along the same latitude of the earth (Lorenz, 2006a).

**Appendix B: Proof of Equation (3)**

This supplement is to prove that, with $K$ members, the ensemble mean and the ensemble spread adjusted by Eq. (3) are the same as those derived with $(K+M)$ members.

With $\bar{\mathbf{v}}_i|_{(K+M)}$ denoting the ensemble mean of the $(K+M)$ ensemble members, we define that $\mathbf{v}_i$ is the $i^{th}$ ensemble member, and $\mathbf{v}'_i$ is the perturbation deviated from $\bar{\mathbf{v}}_i|_{(K+M)}$.

We select the first $K$ members and define the new $i^{th}$ member with Eq. (C1).

$$\mathbf{v}_i^{new} = \bar{\mathbf{v}}_i|_{(K+M)} + \left[\frac{1}{K}\sum_{j=K+1}^{j=K+M}\mathbf{v}'_j + \mathbf{v}'_i\right]\left(\frac{\sigma_{K+M}}{\sigma_K}\right) \tag{C1}$$

In Eq. (C1), $\sigma_{K+M}$ is the standard deviation from the $(K+M)$ members and $\sigma_K$ is the standard deviation of the first $K$ members.

$$\sum_{i=1}^{K}\mathbf{v}_i^{new} = K\bar{\mathbf{v}}_i|_{(K+M)} + \left[\sum_{j=K+1}^{j=K+M}\mathbf{v}'_j + \sum_{i=1}^{i=K}\mathbf{v}'_i\right]\left(\frac{\sigma_{K+M}}{\sigma_K}\right) \tag{C2}$$

Since $\left[\sum_{j=K+1}^{j=K+M}\mathbf{v}'_j + \sum_{i=1}^{i=K}\mathbf{v}'_i\right] = \sum_{i=1}^{i=K+M}\mathbf{v}'_i = 0$, the mean of the new $K$ ensemble members is $\bar{\mathbf{v}}_i|_{(K+M)}$. In addition, $\left(\frac{\sigma_{K+M}}{\sigma_K}\right)$ is the scaling factor to ensure that the new ensemble spread of the $K$ member has the same ensemble spread as the $(K+M)$ members.

**Author contributions**

Yung-Yun Cheng is responsible for all plots, initial analysis, and some writing; Shu-Chih Yang proposes the idea; leads the project, organizes and refines the paper; Zhe-Hui Lin provides the initial framework of the experiments, and Yung-An Lee provides significant discussions and inputs for this research.

**Competing interests**

The authors declare that they have no conflict of interest.

**Acknowledgments**

The authors are very thankful for the valuable discussion with Dr. Jeff Steward. The authors are also very grateful for the
215 initial discussion with Dr. Takemasa Miyoshi from RIKEN. This work is sponsored by the Taiwan NSTC grant 111-2111-M-008-030.

**Code and data availability**

Details of the experiment data could be reproduced by the code. The control run and code will be released at Github (https://github.com/yungyun0721/orthogonal_vector_code) after this manuscript is accepted.

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

**Table 1.** Lists of experiments and their corresponding configurations. The experiments with pseudomembers use the same settings as the standard LETKF, denoted with CNTL in each set of experiments.

| | DA setup | | observation | | LETKF parameters | | | |
|---|---|---|---|---|---|---|---|---|
| | interval (steps) | Inflation | number | Error variance | Ens. size | Truncation Loc. (degrees) | Localization (degree) | pseudovector |
| CNTL | 30 | 1.8 | 20 | 1.0 | 6 | 45 | 12.5 | N |
| Orth_IESV1 | | 1.7 | | | 6+1 | | | IESV1 |
| Orth_EMV | | 1.8 | | | 6+1 | | | EMV |
| Orth_EMV+Orth_IESV1 | 30 | 1.9 | 20 | 1.0 | 6+2 | | | EMV and IESV1 |
| CNTL_OERR2 | | 1.7 | | 2.0 | 6 | | | N |
| Orth_IESV_OERR2 | | 1.7 | | 2.0 | 6+1 | | | IESV1 |
| Orth_EMV_OERR2 | | 1.7 | | 2.0 | 6+1 | | | EMV |
| CNTL_OBS10 | | 1.0 | 10 | | 6 | | | N |
| Orth_IESV_ OBS10 | | 1.4 | | | 6+1 | | | IESV1 |
| Orth_EMV_ OBS10 | | 1.4 | | | 6+1 | | | EMV |
| CNTL_short | 10 | 1.4 | 20 | 1.0 | 6 | 45 | 12.5 | N |
| Orth_IESV_short | | 1.5 | | | 6+1 | | | IESV1 |
| Orth_EMV_short | | 1.4 | | | 6+1 | | | EMV |
| CNTL_long | 50 | 1.7 | 20 | 1.0 | 6 | 45 | 12.5 | N |
| Orth_IESV_long | | 1.8 | | | 6+1 | | | IESV1 |
| Orth_EMV_long | | 1.7 | | | 6+1 | | | EMV |
| M20_CNTL | 30 | 1.4 | 20 | 1.0 | 20 | 90 | 25 | N |
| M20_Orth_IESV1 | | 1.4 | | | 20+1 | | | IESV1 |
| M20_Orth_EMV | | 1.5 | | | 20+1 | | | EMV |
| M20_Orth_IESV1-5 | | 1.8 | | | 20+5 | | | First 5 IESVs |
| M20_Orth_IESV1-10 | | 1.7 | | | 20+10 | | | First 10 IESVs |
| M20_Orth_IESV1-15 | | 1.9 | | | 20+15 | | | First 15 IESV10s |
| M7_LETKF | 30 | 1.8 | 20 | 1.0 | 7 | 45 | 12.5 | N |
| M25_LETKF | 30 | 1.4 | 20 | 1.0 | 25 | 108 | 27 | N |

| M30_LETKF | 30 | 1.1 | 20 | 1.0 | 30 | 144 | 36 | N |

**Table 2.** The mean and standard deviation (STD) of the analysis RMSE of 10 experiments using the standard configuration. Each experiment is initialized with a different random seed. RMSE is only computed for the analysis times when the CNTL analysis RMSE is two standard deviations larger than the mean CNTL analysis RMSE. The number in the bracket indicates the improvement rate with respective to the CNTL RMSE.

| | Mean | STD |
|---|---|---|
| CNTL | 3.781 | 0.207 |
| IESV | 2.011 (46.7%) | 0.168 |
| Orth_IESV1 | 2.048 (45.7%) | 0.235 |
| EMV | 1.949 (48.4%) | 0.210 |
| Orth_EMV | 1.919 (49.1%) | 0.181 |

**Table 3.** RMSE of different experiments.

| | Total DA cycles (550 cycles) | | Large Analysis errors (about 25 cycles) | |
|---|---|---|---|---|
| | Background | Analysis | Analysis | 30-step Forecast |
| CNTL | 2.1847 | 1.6571 | 3.5685 | 3.9456 |
| Orth_IESV1 | 2.1359 | 1.5859 (4.30%) | 1.9096 (46.49%) | 2.5432 (35.54%) |
| Orth_EMV | 2.0895 | 1.5262 (7.90%) | 1.7621 (50.62%) | 2.3585 (40.22%) |
| CNTL_2 | 2.6451 | 2.0730 | 3.6482 | 3.3842 |
| Orth_IESV_2 | 2.6430 | 2.0521 (1%) | 2.6177 (28.25%) | 3.1114 (8.06%) |
| Orth_EMV_2 | 2.5630 | 1.9981 (3.61%) | 2.5392 (30.40%) | 2.9690 (12.27%) |
| CNTL_OBS10 | 4.0475 | 4.4914 | 4.6066 | 4.1115 |
| Orth_IESV_OBS10 | 3.8944 | 4.1480 (7.65%) | 3.0303 (34.22%) | 3.3529 (18.45%) |
| Orth_EMV_OBS10 | 3.7956 | 3.9999 (10.94%) | 3.5619 (22.68%) | 3.6218 (11.91%) |
| CNTL_short | 0.7903 | 0.6713 | 1.1369 | 1.3288 |
| Orth_IESV_short | 0.7269 | 0.6169 (8.10%) | 0.8473 (25.47%) | 0.9950 (25.12%) |
| Orth_EMV_short | 0.7101 | 0.6012 (10.44%) | 0.8796 (22.63%) | 1.0477 (21.15%) |
| CNTL_long | 3.3472 | 2.8679 | 4.6489 | 3.6933 |
| Orth_IESV_long | 3.3102 | 2.8104 (2.00%) | 3.0725 (33.90%) | 3.3123 (10.32%) |
| Orth_EMV_long | 3.2621 | 2.7292 (4.84%) | 3.3590 (27.75%) | 3.4358 (6.97%) |
| CNTL_M20 | 1.6483 | 1.0799 | 2.4594 | 2.7106 |
| M20_Orth_IESV1 | 1.5959 | 1.0482 (2.94%) | 1.5810 (35.72%) | 2.1618 (20.25%) |
| M20_Orth_EMV | 1.6034 | 1.0571 (2.11%) | 1.8175 (26.10%) | 2.2292 (17.76%) |
| M20_Orth_EMV+IESV1-4 | 1.6509 | 1.0714 (0.79%) | 1.4492 (41.08%) | 1.9708 (27.30%) |
| M20_Orth_IESV1-5 | 1.6519 | 1.0797 (0.02%) | 1.4978 (39.10%) | 1.9740 (27.17%) |
| M20_Orth_IESV1-10 | 1.5540 | 1.0431 (3.41%) | 1.3430 (45.39%) | 1.7390 (35.84%) |
| M20_Orth_IESV1-15 | 1.5013 | 0.9752 (9.70%) | 1.2637 (48.62%) | 1.7363 (35.94%) |

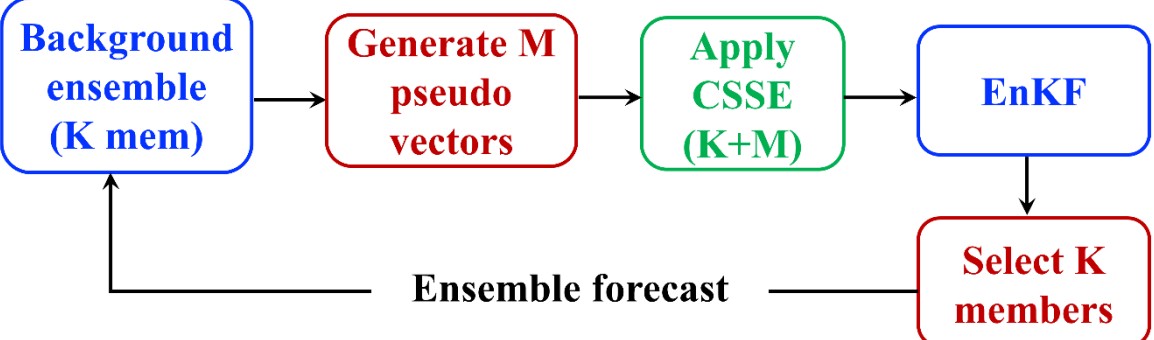

**Figure 1: The framework of the increased-size EnKF system**

# Local max (F-T) area (7grids) in average

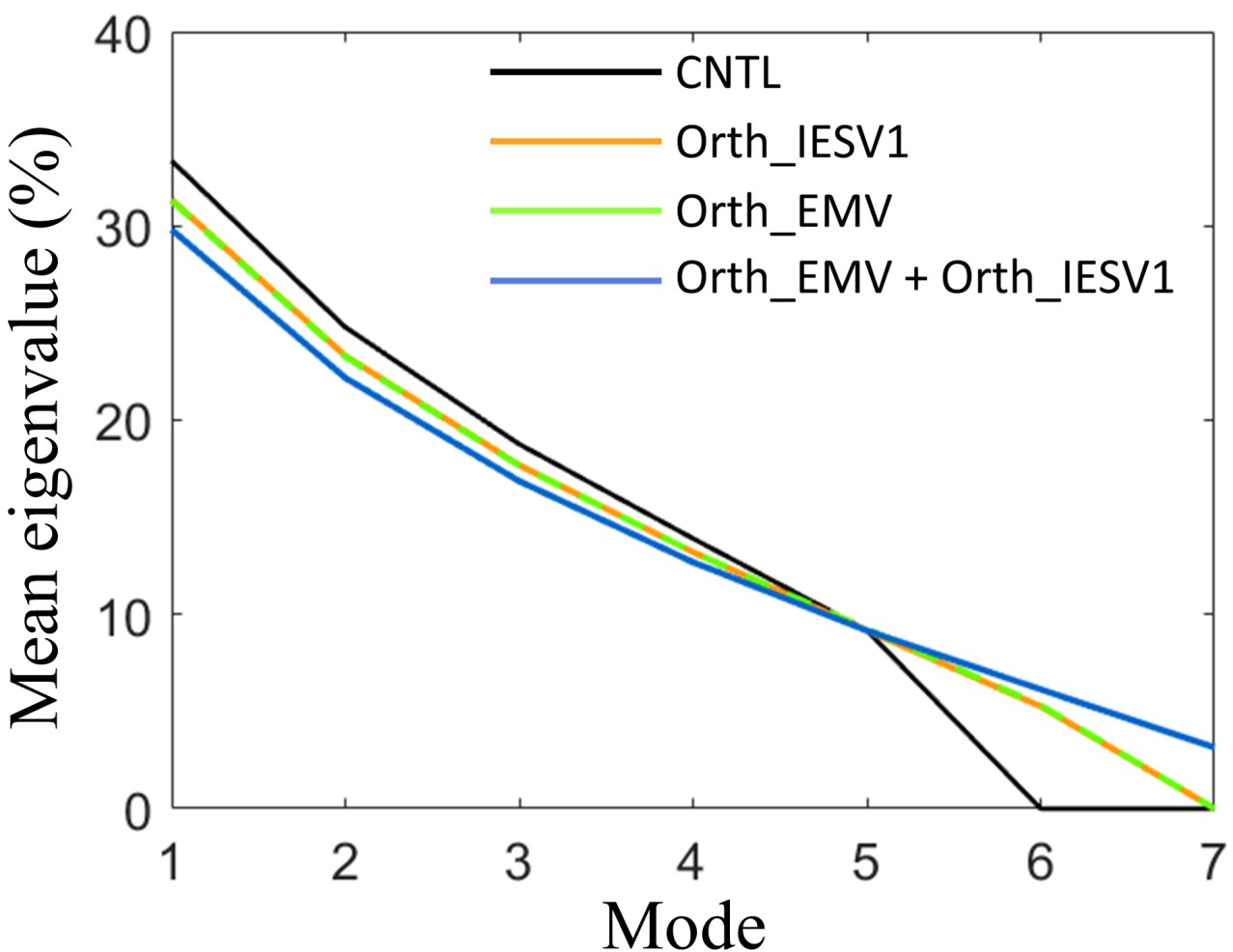

**Figure 2: The mean eigenvalue percentage (Y-axis) of the control run (black), the average RSV (gray), the orthogonal IESV1 (orange), the orthogonal EMV (green), and the 8-member experiments (blue) in each eigenmode (X-axis). The calculation is done using the ensemble perturbation in the local area centered at the grid with the maximum forecast error.**

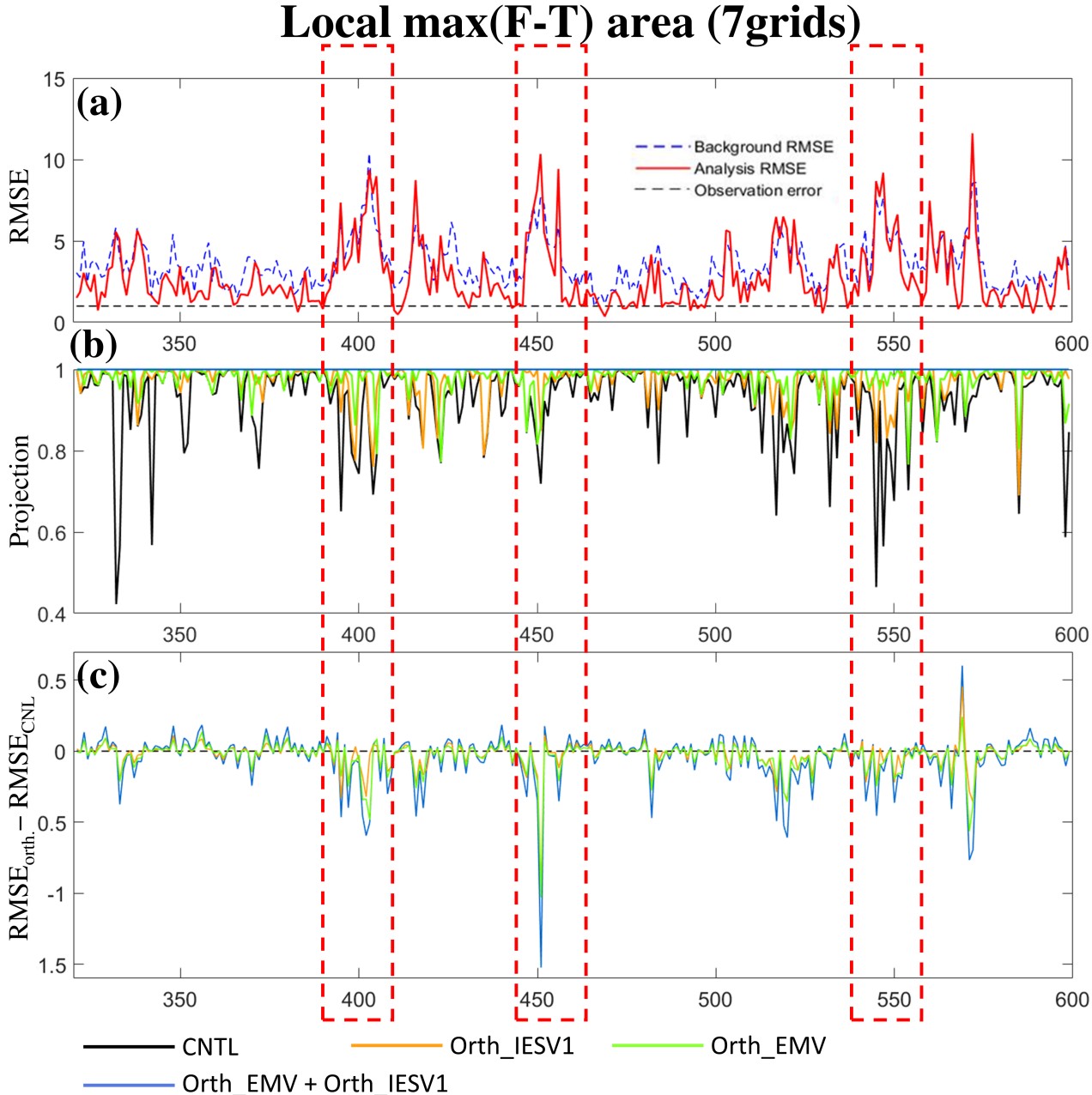

**Figure 3:** Time series of (a) the background (blue dashed line) and analysis (red) RMS errors, (b) the projection on the background error, and (c) the RMSE differences between experiments using pseudomember and CNTL (negative indicates improvement).

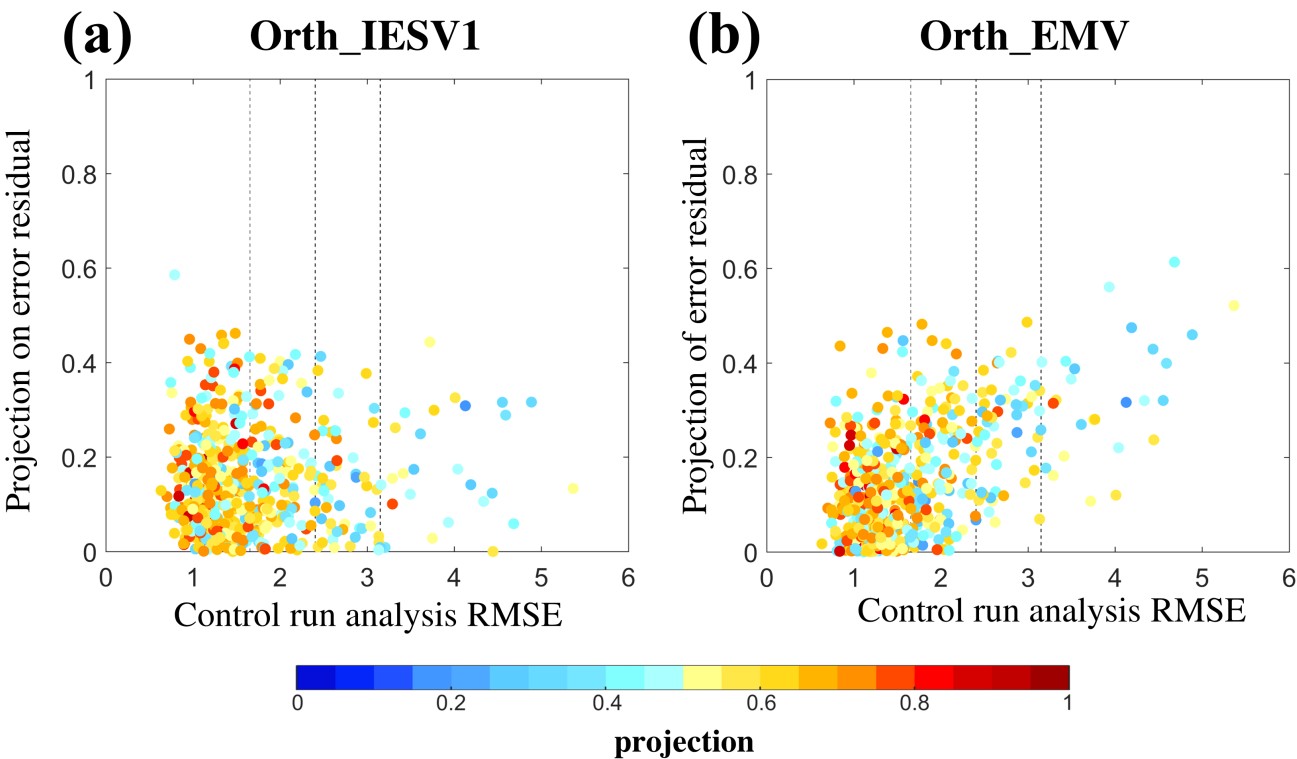

**Figure 4: Scatterplots of the projection on the background error (shading) according to the control analysis RMSE (X-axis) and the projection of the added pseudomember on the forecast errors residual: (a) orthogonal IESV1 and (b) orthogonal EMV experiments.**

280

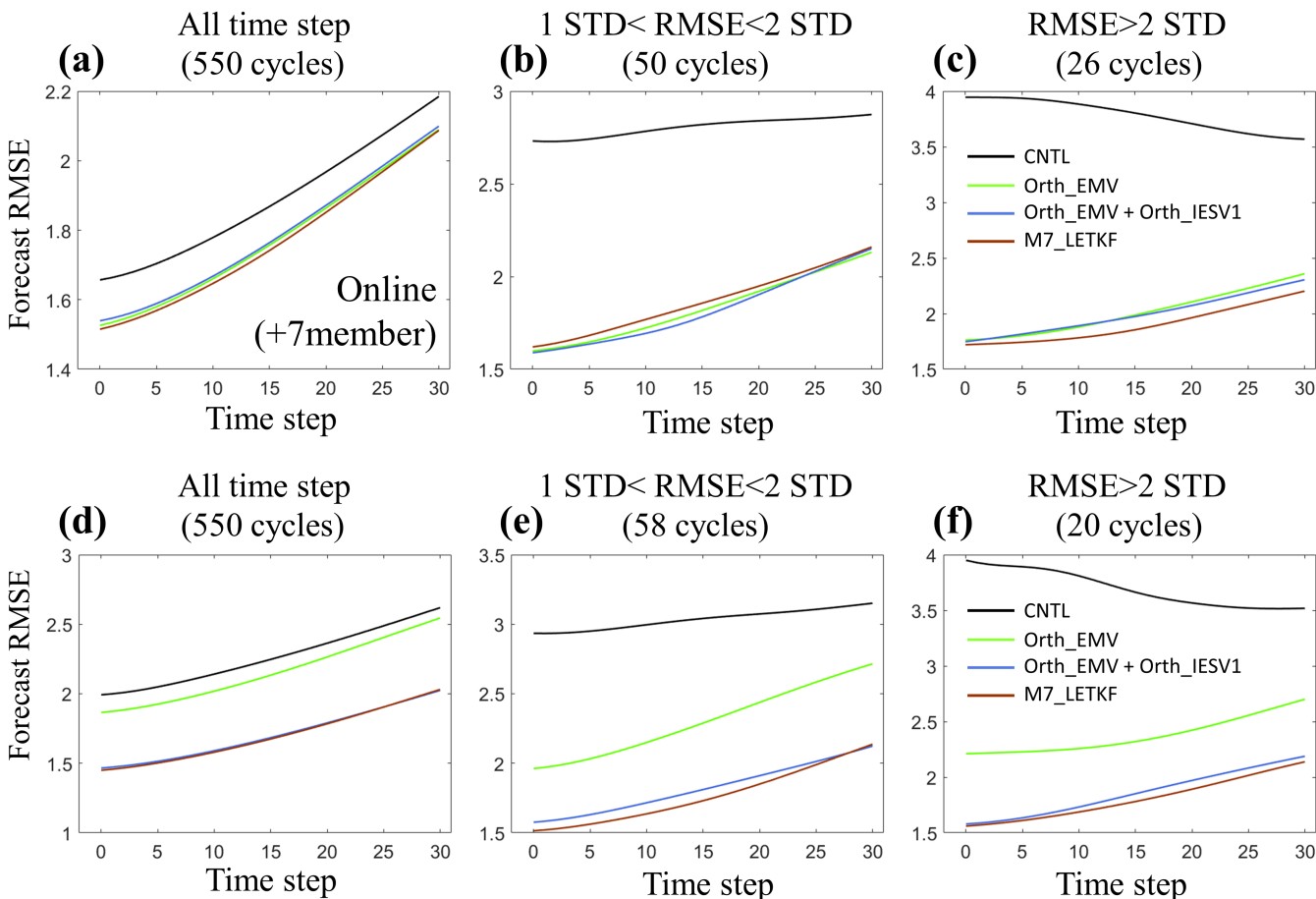

**Figure 5: Mean analysis and forecast RMS errors (Y-axis) during the 30-step integration (X-axis) with different experiment. (a)-(c) the perfect model experiment and (d)-(e) the imperfect model experiment. (a) and (d) are averaged for all analysis cycles. (b) and (e) uses the cycles when the CNTL analysis RMSE values are between one and two standard deviations larger than the mean RMSE. (c) and (f) are uses the cycles when the CNTL analysis RMSE values are two standard deviations larger than the mean RMSE. In the imperfect model experiments, the *F* value in the governing equations of the Lorenz-96 model is changed from 8 to 7.8, and the inflation increases to 2.3.**

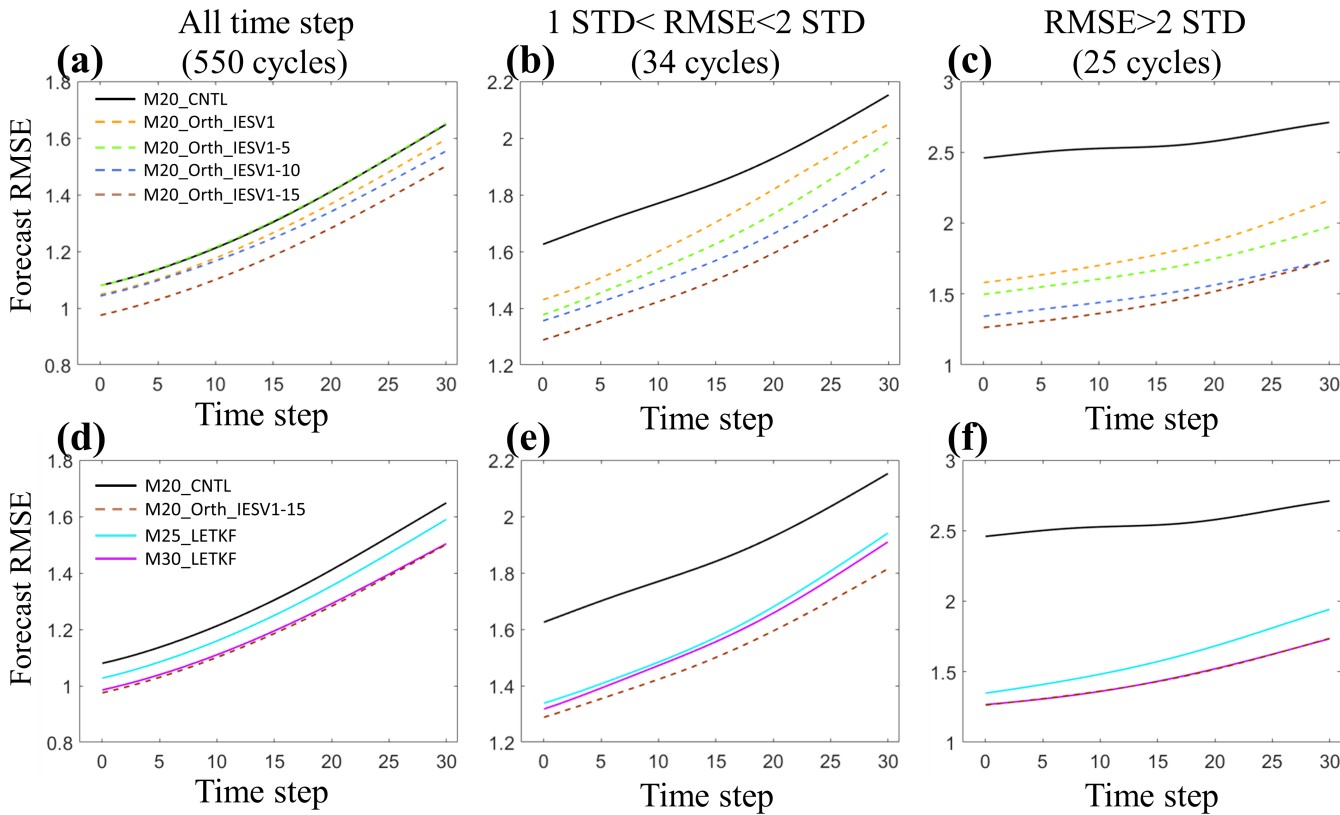

**Figure 6: The same as Fig. 5, expect for different experiments. CNTL_M20, M25-mem_LETKF and M30_LETKF use the standard LETKF with 20, 25 and 30 members, respectively. M20_Orth_IESV1, M20_Orth_IESV1-5, M20_Orth_IESV1-10 and M20_Orth_IESV1-15 use 20 members and orthogonal components of IESVs as the pseudomembers.**