# Peer review of "Using orthogonal vectors to improve the ensemble space of the EnKF and its effect on data assimilation and forecasting"

_Nonlinear Processes in Geophysics, 2022_

## Referee Comment (RC1)

**Review report for npg-2022-19 "Using orthogonal vectors to improve the ensemble space of the EnKF and its effect on data assimilation and forecasting"**

**Main concerns**

The idea of improving the performance of an EnKF by artificially increasing the ensemble size appears sensible in general. Nevertheless, there appears to be a major weakness in the current manuscript, that is, the technical idea is not fully supported/explained by the numerical experiments, and I think more thorough numerical investigations would be required. In this regard, I'd like to invite the authors to consider the following questions/suggestions:

1. If it's beneficial to artificially increase the ensemble size, why did the authors only consider to increase the size by one or two in their implementation? In general, how many more pseudo-members can the authors include before the EnKF's performance saturates (or perphas deteriorates)?

2. The authors considered a few different ways (EMV, IESV, RSV) to add ensemble members and concluded that EMV appears to be better. Any reason to explain the better performance of EMV?

3. There are many factors, such as inflation factor, frequency and density of the observation operator, the variance of observation errors, the length of the assimilation time window and even random seeds, which could potentially affect the performance of an EnKF. I's suggest that the authors conduct an extensive investigation on the impacts of these factors on the performance of the EnKF.

4. In particular, I'd suggest the authors repeat one experiment for a number of times (each time with a different random seed) to reduce the effect of statistical fluctuations. In the current manuscript, I got the impression that the authors did each experiment only once, which does not make much sense for the L96 model, since it's quite cheap to run. If the same experiment is repeated multiple times, then I'd further suggest that the authors use a table

to summarize the results (RMSE) in the form of mean $\pm$ STD, which would make the performance comparison statistically more meaningful.

**Minor issues**

1. L18, P1: The statement "The BEC is sampled by the background ensemble" does appear precise, since one cannot sample a covariance (perhaps you want to say "estimate" instead?)

2. L22, P1: In linear algebra, the statement "cause the ensemble space to be underestimated" also appears problematic. In which sense is a (linear) space underestimated? My guess is that here by "ensemble space" the authors possibly mean ensemble variance instead. If so, it seems that there are many similar instances later in which the terminology "ensemble space" is not correctly used (e.g., as in "apply SVD to the ensemble space" in L63, P3).

3. The authors stated that using Eq. 3 can preserve the ensemble mean and spread. It'd be good to provide a proof for this statement.

4. L92, P3: Rephrase the text "as a demonstration of proof of concept".

5. In Caption of Figure 4, "according" is duplicated.

---

## Author Comment (AC1)

**Response to Reviewer #1**

Dear Reviewer,

We would like to thank the reviewer's instrumental suggestions, which have greatly improved our manuscript. We have conducted new experiments to fully explore the performance of the new method. Our manuscript is significantly revised to include the results of new experiments and to address your comments and suggestions. Please see our point-by-point responses to your comments as follows.

**Major comments:**

1. If it's beneficial to artificially increases the ensemble size, why did the authors only consider to increase the size by one or two in their implementation? In general, how many more pseudo-members can the authors include before the EnKF's performance saturates (or perhaps deteriorates).

Thank you very much for this important comment. We have conducted new experiments with a large ensemble size (20 members). As shown in the new Table 3, the benefit of adding pseudomembers based on IESVs increases as the number of pseudomember increases, particularly when the standard LETKF (M20_CNTL) has large analysis RMSE. Using more than 10 IESVs can reduce the analysis RMSE by more than 45% and the IESV1 (M20_Orth_ESV1) provides the dominant effect. The improvement rate with 15 IESVs becomes saturated for the following 30-step forecast. Such a performance (M20_Orth_ESV1-15) is better than the 25-member standard LETKF in general, and even better than the 30-member standard LETKF for the group of mildly large analysis errors.
We have included the following figure (new Figure 6) and relevant discussion at lines 155-159 in the revised manuscript.

[Figure]

Figure: The mean RMSE of the forecast error during the 30-step integration. CNTL_M20, M25-mem_LETKF and M30_LETKF use the standard LETKF with 20, 25 and 30 members, respectively. M20_Orth_IESV1, M20_Orth_IESV1-5, M20_Orth_IESV1-10 and M20_Orth_IESV1-15 use 20 members and orthogonal components of IESVs as the pseudomembers.

2. The authors considered a few different ways (EMV, IESV, RSV) to add ensemble members and concluded that EMV appears to be better. Any reason to explain the better performance of EMV?

In dynamic systems, the ensemble perturbations evolve upon the mean structure. In other words, the dynamic errors are expected to relate to the instabilities existing in the ensemble mean. Therefore, the structure of the ensemble mean is effective in expanding the ensemble space to capture the error. A better performance with EMV as the pseudomemer suggests that the background error also projects on the structure of ensemble mean. We observed that EMV is particularly useful when the ensemble size is small. With a small ensemble size, the ensemble can not capture the forecast error well and thus EMV is particularly useful. With a large ensemble size (and longer localization), the large-scale error in the background mean state is much reduced and the structure of the mean state is less effective as the pseudomember, compared with IESV1. We have provided the explanation at lines 168-170.

3. There are many factors, such as inflation factor, frequency and density of the observation operator, the variance of observation errors, the length of the assimilation time window and even random seeds, which could potentially affect the performance of an EnKF. I's suggest that the authors conduct an extensive investigation on the impacts of these factors on the performance of the EnKF.

We agree that we should investigate the performance of the newly proposed method under different assimilation configurations. To address this comment, we have carried out a series of experiments to investigate how the new framework is sensitive to assimilation frequency, density of observation, assimilation interval and random seeds.

Our conclusion remains under all different assimilation configurations, but with different degrees of improvement rate. Therefore, we can conclude that adding pseudomembers can effectively improve the analysis accuracy, in particular when the standard LETKF performs poorly. We note that the inflation factors used in all experiments are optimized.

Please see the new Table 3 and the relevant discussions section 3.2 in the revised manuscript.

4. In particular, I'd suggest the authors repeat one experiment for a number of times (each time with a different random seed) to reduce the effect of statistical fluctuations. In the current manuscript, I got the impression that the authors did each experiment only once, which does not make much sense for the L96 model, since it's quite cheap to run. If the same experiment is repeated multiple times, then I'd further suggest that the authors use a table to summarize the results (RMSE) in the form of mean +/- STD, which would make the performance comparison statistically more meaningful.

Following the reviewer's suggestion, we have repeated one standard experiment 10 times with different random seeds to derive statistically meaningful results.

We summarized the RMSE of the experiments in Table 2 in the form of mean and STD. It is clear the EMV is effective in reducing the analysis RMSE when the standard LETKF (CNTL) has large analysis errors and using the orthogonal component of EMV shows a better performance than EMV. However, adding the orthogonal component of IESV1 as the pseudomember results in the largest RMSE STD among the experiments. Even when the orthogonal IESV1 is sometimes less effective, it is still useful in improving the analysis accuracy (a smaller mean RMSE than that of CTRL). Please see the discussion at lines 130-141.

**Specific comments:**

1. L18, P1: The statement "The BEC is sampled by the background ensemble" does appear precise, since one cannot sample a covariance (perhaps you want to say "estimate" instead?)

Thank you. We have replaced "sampled" to "estimated" for this sentence.

2.  L22, P1: In linear algebra, the statement "cause the ensemble space to be underestimated" also appears problematic. In which sense is a (linear) space underestimated? My guess is that here by "ensemble space" the authors possibly mean ensemble variance instead. If so, it seems that there are many similar instances later in which the terminology "ensemble space" is not correctly used (e.g., as in "apply SVD to the ensemble space" in L63, P3).

Thank you for your suggestion. We have modified this sentence to the use of finite ensemble size can cause underestimation of the background error variance and the background error correlation is less optimally represented due to the sampling error.

We have also revised the sentence at line 63 to emphasize that "SVD is applied to find the orthogonal vectors to represent the space spanned by the ensemble members." We also clarified the sentences using the ensemble space (e.g. lines 68, 93, 104).

3.  The authors stated that using Eq. 3 can preserve the ensemble mean and spread. It'd be good to provide a proof for this statement.

Appendix B is provided as a proof for the statement of Eq. 3.

4.  L92, P3: Rephrase the text "as a demonstration of proof of concept".

This sentence is removed from the revised manuscript.

5.  In Caption of Figure 4, "according" is duplicated.

Sorry for the typo. We have corrected it.

---

## Author Comment (AC2)

**Response to Reviewer #2**

Dear Reviewer,

We would like to thank the reviewer's instrumental suggestions, which have greatly improved our manuscript. We have conducted new experiments to fully explore the performance of the new method. Our manuscript is significantly revised to include the results of new experiments and to address your comments and suggestions. Please see our point-by-point responses to your comments as follows.

**Major comments:**

1. It would be beneficial to conduct more experiments and include their results and analysis of the results in the manuscript to demonstrate how the use of this method affects data assimilation. In particular, I would recommend exploring how varying localization, observation density and observation errors affect the results, since all of those parameters influence the "effective observation dimension" (the degrees of freedom that are required for assimilating observations (Kirchgessner et al, 2014)).

Thank you very much for this comment. We have conducted more experiments to investigate how the performance of this method is sensitive to different assimilation configurations, including observation density, observation errors, assimilation intervals and ensemble size.

2. The results presented in Figure 6 suggest a bigger benefit from using N+1=7 ensemble members in DA vs using N ensemble members + 1 pseudomember. While I agree that using pseudomember vs ensemble member saves on running an ensemble forecast, the savings are only N/(N+1) for the ensemble forecast runs. It would be interesting to see if there are benefits of using more than one or two ensemble pseudomembers so the savings on not running extra ensemble forecasts may be more justifiable.

Thank you very much for this important comment. We have conducted a new series of experiments using 20 ensemble members to investigate how much computation can be saved by this method, and whether the benefit will be saturated with more pseudo-members. As shown in the new Table 3, the benefit of adding pseudomembers based on IESVs increases as the number of pseudomember increases. When the standard LETKF (CNTL_M20) has large analysis RMSE, using more than 10 IESVs can reduce the analysis RMSE by more than 45% and the IESV1 (M20_Orth_IESV1) provides the dominant effect. The improvement rate with 15 IESVs saturates for the following 30-step forecast. Such a performance (M20_Orth_IESV1-15) is better than the 25-member standard LETKF in general and even better than the 30-member standard LETKF for the group of mildly large analysis errors. The forecast computation is only 66% of the 30-member LETKF.

We have included the following figure (new Figure 6) and relevant discussion at lines 154-159 in the revised manuscript.

[Figure]

Figure: The mean RMSE of the forecast error during the 30-step integration. CNTL_M20, M25-mem_LETKF and M30_LETKF use the standard LETKF with 20, 25 and 30 members, respectively. M20_Orth_IESV1, M20_Orth_IESV1-5, M20_Orth_IESV1-10 and M20_Orth_IESV1-15 use 20 members and orthogonal components of IESVs as the pseudomembers.

3. Experiments in section 3.2 include analysis (in Figure 5) of results when IESV1 and EMV are used with and without orthogonalization. Please include description of what the difference between those are.

Figure 5 in the previous manuscript is removed. We now compare the results with Table 2. When IESV1 and EMV used with orthogonalization means that we took the component of IESV (or EMV) orthogonal to the ensemble space. We have clarified the differences between those experiments at lines 73-74.

4. Please describe what "average RSV" means for the experiments in section 3.2 (Figure 5). From Figure 5 it appears that results with "average RSV" are similar to results with orthogonal IESV1. It may be good to include the statistical significance of the differences between the experiments and also discuss this particular result.

Due to the limitation of the manuscript length, we removed the results of RSV in the revised manuscript. To show the significance of the new method, we have repeated the standard experiment 10 times (each time with a different random seed). The results of RMSE are summarized in Table 2 in a form of mean and STD.

With the 10 randomly initialized standard experiments, adding IESV1 or EMV as the pseudovector is always effective in improving the CNTL analysis. When the CTRL has large analysis errors, both IESV1 and EMV can even reach a mean improvement rate larger than 46%. On average, the Orth_EMV show a larger mean improvement than the EMV.

Please see our discussions at line 134-141.

**Specific/inline comments:**

1. Figure 5: the gray line is very hard to see, I suggest changing the color.
We apologized for the unclear line. Figure 5 is now removed from the revised manuscript.

2. Figure 3c: it is very hard to see the distinction between different experiments, it may be good to include statistics of the error differences.
Following the reviewer's suggestion, the statistics of the error are included in Table 3 with improvement rates. We also replot Figure 3c to better illustrate the different experiments.

---

## Author Response (AR2)

Dear Editor,

Thank you very much for your suggestions. We have revised our manuscript accordingly. Please see our point-by-point response as follows.

**1) I did not find your response to Rev. 1's 2nd major comment about the reason for the good performance of EMV vectors completely clear or convincing. Also, I could not find a related explanation in the text at lines 168-170. Could you please clarify your response and update the text of the manuscript if necessary? If your explanation is tentative or speculative, please say so.**

We apologize for the unclear explanation about the good performance of EMV.
Our results suggest that using the EMV as the pseudomember is very effective when the ensemble size is small. With the small ensemble size, the ensemble space spanned by the ensemble perturbations cannot capture the background error well, and the corrections for the background mean are less optimal. We speculate that this limitation is more evident for the large-scale error due to using a small localization with a small ensemble size. As a result, the structure of the ensemble mean projects on the background error. Figure 4 shows that, the orthogonal EMV projects more onto the forecast error residual at most analysis times, compared to the orthogonal IESV1. Therefore, the orthogonal EMV is more useful to increase the ensemble space to reduce the analysis errors (a better performance). With a large ensemble size and a large localization (Table 1), the large-scale error in the background mean state is much reduced, and the structure of the mean state is less effective in being used as the pseudomember.

The explanations are provided at lines 121-124 and 160-166.

**2) I suggest you drop "Results" from subtitles 3.1 and 3.2, given Section 3 is entitled Results**

Thank you for your suggestions. We have revised our manuscript accordingly.